# MSCFFN: A New FFN with Multi-Space Cross to Accelerate Transformer

**Dongge Tang**
Beihang University, Beijing, China
Du Xiaoman Financial, Beijing, China
tangdongge@buaa.edu.cn

**Qing Yang**
Du Xiaoman Financial, Beijing, China
yangqing@duxiaoman.com

## Abstract

Transformer models have achieved impressive success in various natural language processing tasks. But it is also limited used in some areas and the heavy computation complexity is one of the main limitations. Many model structures have been proposed to reduce the computation complexity and some are really effective. The previous research can be divided into two categories. One is to use more effective training and inference strategies and the other is focused on how to replace the standard self-attention mechanism with linear attention method. Differently, we revisit the design in Transformer and find that the feed forward network (FFN) is also computationally expensive, especially when the hidden dimension is large. In this paper, we propose a new FFN structure, named MSCFFN, which splits the large matrix space to several small space to reduce the computation complexity and uses the Multi-Space Cross method to ensure the accurate result. To the best of our knowledge, this is the first time to redesign FFN to accelerate Transformers. We experimentally validate the effectiveness of the proposed method on the Long-Range Arena benchmark. And the results show MSCFFN can achieve a faster speed with a similar or even better accuracy. Our codes are available at https://github.com/henriTang/Transformer_Speed.

## 1   Introduction

Since Transformer models (Vaswani et al., 2017) were proposed, especially the Bert (Devlin et al., 2018) with pre-trained language models, they have achieved state-of-the-art results in various domains of natural language processing, such as machine translation (Liu et al., 2020; Bao et al., 2021), text classification (Xiong et al., 2021; Yang et al., 2020), natural language inference (Jiang and de Marneffe, 2019; Guo et al., 2019), question answering (Yang et al., 2019; Heo et al., 2022) and so son.

The effectiveness of transformer comes from the multi-head self-attention mechanism (MHSA), fully connected feed-forward network (FFN) and the deep layers. The MHSA helps model to learn the relation between tokens in different position of a sequence, the FFN helps enlarge the network space and enhance the non-linear capability, and the deep layers are helpful to build a more complex sentence relationship. However, these operations also cause the high computation complexity, and it is hard to use for user with low computation resource. To tackle this essential obstacle, there has been much prior literature on improving the efficiency of Transformers. The main research consists of two parts, one is to use more effective strategies. For example, Pyramid-BERT (Huang et al., 2022) uses the core-set based token selection method to reduce complexity, and PCEE-BERT (Zhang et al., 2022) accelerates BERT using patient and confident early exiting method. The other is focused on how to reduce the computation complexity of the standard self-attention, which is quadratic complexity correlated with sequence length, to linear attention method. Such as Performer (Choromanski et al., 2020), Longformer (Beltagy et al., 2020), Cosformer (Qin et al., 2022), Flowformer (Wu et al., 2022) and so on.

Although these methods have been proven to be very effective in reducing computation complexity, they almost don't discuss the influence of hidden dimension to computation complexity. In fact, transformer models are also quadratic complexity correlated with hidden dimension. When the hidden dimension is large, it is computationally prohibitive to train or inference (Zhang and Cai, 2022). Actually, to achieve better results, Bert xlarge uses 2048 of hidden dimension and Albert xxlarge (Lan et al., 2019) uses 4096 of hidden dimension, which is really computationally expensive for a common researcher. For the transformer models, the main unit with quadratic complexity of

hidden dimension ($d$) is FFN which has the complexity of $O(8d^2)$ in vanilla transformer, because the dimension of inner-layer is usually set to $4d$ for a better result. Therefore, in this paper, we focus on how to reduce the computation complexity of FFN.

According to the characteristic of quadratic complexity related to hidden dimension, if we shrink the hidden dimension to $\frac{1}{n}d$, the computation complexity will be reduced to $\frac{1}{n^2}$ of the original. Therefor, we firstly split the original large matrix space to several matrix spaces, and we apply the different weight parameters for each small matrix to conduct the feed-forward network separately. Obviously, the representation ability of n small matrices with $\frac{d^2}{n^2}$ which is the space of weight parameters is much less than it of $d^2$, and it is easy to cause the effect decline. In order to solve this problem, we propose a method of space cross mechanism, and we name the final model MSCFFN. Through the cross of different small space, we can enlarge the matrix space and obtain a stronger representation capability. Furthermore, benefitting from the information from different representation subspaces, a larger coefficient of inner-layer and the deeper network of FFN, MSCFFN can achieve a better accuracy with a faster speed. We conduct experimental study on the Long-Range Arena benchmark (Tay et al., 2020), and the results demonstrate that our MSCFFN is effective in speeding up and performs better than baselines.

## 2 Related work

As mentioned in section 1, The most common methods to reduce the computation complexity are as follows:

Effective strategies: with this method, they don't change the structure of MHSA and FFN, but change the strategies of layers to train or tokens to train. For example, (Dai et al., 2020) proposed Funnel-Transformer which gradually compresses the sequence of hidden states to a shorter one and hence reduces the computation cost. (Huang et al., 2022) proposed Pyramid-BERT which used a coreset based token selection method to successively shorten sequence length and void expensive computation resource. PCEE-BERT (Zhang et al., 2022) was proposed by Zhang et al., and it achieves the goal of speedup through exiting early when enough numbers of consecutive intermediate layers are confident about their prediction. GroupBERT (Ch-

elombiev et al., 2021) uses grouped transformations to reduce the density of fully connected layers. Sparse mixtures of experts (sparse MoEs) reduce the computation cost through conditional computation which only activates a subset of the overall model, such as (Zhang et al., 2021; Fedus et al., 2022).

Linear attention method: This method is to resolve the quadratic complexity of vanilla self-attention mechanism and try to reduce the complexity of $O(n^2)$ to $O(nlog(n))$ or even $O(n)$. For example, Locally-sensitive hashing (LSH) attention (Kitaev et al., 2020) applies a multi-round hashing scheme to compute dot-product attention. Longformer utilizes a sliding windows to learn the token relation in a local window. Performer introduces an unbiased linear estimation of the attention matrices. Cosformer proposes a linear attention by using the kernel methods with a non-linear re-weighting scheme.

## 3 Method

### 3.1 Preliminaries of vanilla Transformers

In this paper, we only talk about the encoder structure, but our model is universal for all FFN. The Transformers is composed of a stack of $N$ identical layers, and each layer contains a MHSA and a FFN. Let us use $X_s \in R^{l \times d}$ and $X_f \in R^{l \times d}$ to represent the input of MHSA and FFN respectively, in which $l$ is the sequence length and $d$ is the hidden dimension.

For the MHSA, the input is firstly linearly projected to $Q \in R^{n \times d}, K \in R^{n \times d}, V \in R^{n \times d}$ by the parameter matrices $W^Q \in R^{d \times d}, W^K \in R^{d \times d}, W^V \in R^{d \times d}$:

$$(Q, K, V) = (X_s W^Q, X_s W^K, X_s W^V) \quad (1)$$

And then the attention can be computed as:

$$Attention(Q, K, V) = softmax(\frac{QK^T}{\sqrt{d_k}})V \quad (2)$$

where $d_k$ is the dimension of $Q$ and $K$. After that, the attention will be linearly transformed to the output by the parameter matrices $W^O \in R^{d \times d}$. And a residual connection with layer normalization will be added. The output is also the input of FFN:

$$X_f = LN(Attention(Q, K, V)W^O + X_s) \quad (3)$$

For the FFN, two linear transformations with a

ReLU activation will be implemented as:

$$FFN(X_f) = max(0, X_f W_{f1} + b_{f1})W_{f2} + b_{f2} \tag{4}$$

where $W_{f1} \in R^{d \times d_i}$ and $W_{f2} \in R^{d_i \times d}$ are weight parameters. $b_{f1} \in R^{d_i}$ and $b_{f2} \in R^d$ are bias parameters. After that a residual connection with layer normalization will be added.

$$O_X = LN(FFN(X_f) + X_f) \tag{5}$$

From the above equations, we can calculate and obtain the computation complexity of MHSA in each layer is approximately equal to $O(4ld^2 + 2l^2d)$, the computation complexity of FFN in each layer is approximately equal to $O(8ld^2)$ and the total computation complexity in each layer is approximately equal to $O(12ld^2 + 2l^2d)$. The ratio of FFN to MHSA equals $\frac{4d}{d+l}$. Of course, when $l >> d$, the complexity of FFN is not worth considering. But in many cases, this condition is not true, and even $l << d$, such as the daily dialogue system, real-time machine translation and so on. Therefore, optimizing the structure of FFN is really meaningful for reducing the computation complexity. What's more, for the improved linear attention mechanism, the FFN occupies a large proportion no matter how long the sequence is. Our MSCFFN can be combined with linear attention and speedup the transformer models further.

### 3.2 MSCFFN Model

In this part, we present the proposed MSCFFN model in detail.

The structure of MSCFFN is shown in Figure 1. First of all, we transfer linearly the input of FFN to a new space by the parameter $W_{f3} \in R^{d \times d}$ and $b_{f3} \in R^d$ so that we can split it into $n$ subspaces easily.

$$I = X_f W_{f3} + b_{f3} = Concat[I_1, I_2, ..., I_n] \tag{6}$$

where $I_i \in R^{l \times \frac{d}{n}}$ is the $i-th$ matrix in subspace. And then we use $n$ different parameters to transformer the $I_i$ to a larger space.

$$L_i = I_i W_{Ii} + b_{Ii} \tag{7}$$

In this equation, $W_{Ii} \in R^{\frac{d}{n} \times \frac{md}{n}}$ and $b_{Ii} \in R^{\frac{md}{n}}$ are the parameters in which $m$ indicates the coefficient of inner-layer. To make the network easier to learn, the inner-layer usually has a bigger dimension and in vanilla FFN $m$ is set to 4. In our work,

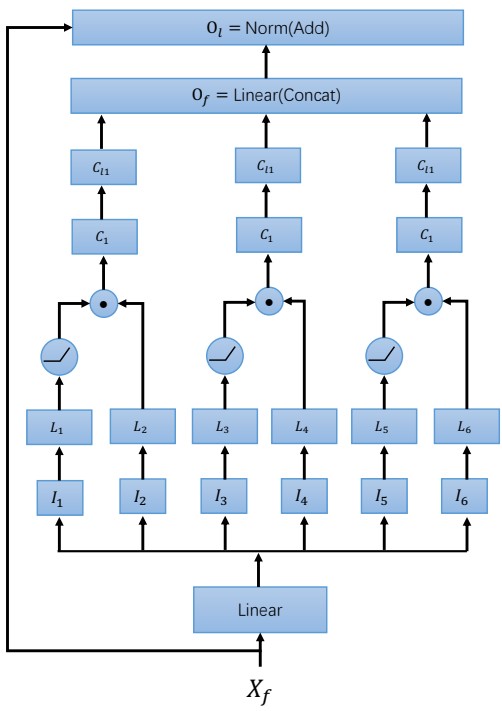

Figure 1: General framework of MSCFFN (here $n = 6$)

we set m to 6 to achieve a better result. We have to admit that the sum of $n$ subspace of $R^{\frac{d}{n} \times \frac{md}{n}}$ is smaller than the original space of $R^{d \times d}$, so the cross method of subspace is proposed to improve the range of representation space. Firstly, The input $L$ is divided into pairs. Secondly, we use non-linear method to one of pairs. Finally, the element-wise product is applied to one with the other in pairs.

$$C_i = Relu(L_{2i-1}) \odot L_{2i} \tag{8}$$

After that a linear transformation is used to project from high dimension to low dimension. And we concat the results to map it linearly to the original dimension of $X_f$.

$$C_{Li} = C_i W_{Ci} + b_{Ci} \tag{9}$$
$$O_f = Concat(C_{Li})W_{CL} + b_{CL} \tag{10}$$

where $W_{Ci} \in R^{\frac{md}{n} \times \frac{d}{n}}$ and $W_{CL} \in R^{\frac{d}{2} \times d}$. Finally, a residual connection followed by a layer normalization is applied as the original FFN.

### 3.3 Theoretical Analysis for MSCFFN

Effectiveness of speed: Through equations above, we can calculate the theoretical computation complexity of MSCFFN which equals $O((1.5 + \frac{2m}{n})ld^2)$. Usually, we set m to 6 and n to 12, so the theoretical computation complexity is $O(2.5ld^2)$.

| Model | ListOps | Text | Retrieval | Image | Pathfinder | Avg |
|---|---|---|---|---|---|---|
| Performer | 18.01 | 65.40 | 53.82 | 42.77 | 77.05 | 51.41 |
| Longformer | 35.63 | 62.85 | 56.89 | 42.22 | 69.71 | 53.46 |
| Transformer | 36.37 | 64.27 | 57.46 | 42.44 | 71.40 | 54.39 |
| Flowformer | 38.70 | 64.29 | 62.24 | 43.20 | 73.95 | 56.48 |
| Performer+MSCFFN | 19.67 | 66.21 | 56.82 | 42.77 | 76.83 | **52.46** (+1.05) |
| Longformer+MSCFFN | 36.70 | 63.61 | 57.59 | 41.26 | 70.25 | **53.88** (+0.42) |
| Transformer+MSCFFN | 37.25 | 64.51 | 60.32 | 43.61 | 71.58 | **55.45** (+1.06) |
| Flowformer+MSCFFN | 38.65 | 64.74 | 61.98 | 43.96 | 74.26 | **56.72** (+0.24) |

Table 1: Results on the Long-Range Arena.

| Model | ListOps | Text | Retrieval | Image | Pathfinder | Avg |
|---|---|---|---|---|---|---|
| Flowformer+MSCFFN | 38.65 | 64.74 | 61.98 | 43.96 | 74.26 | **56.72** |
| Flowformer+FFN$_{low-di}$ | 36.75 | 63.32 | 60.47 | 42.12 | 71.48 | 54.83 (-1.89) |
| Flowformer+FFN$_{S-MoE}$ | 37.59 | 64.16 | 61.18 | 42.91 | 72.96 | 55.76 (-0.96) |

Table 2: Comparing multi-space cross method with FFN$_{low-di}$ and FFN$_{S-MoE}$. The FFN$_{low-di}$ indicates the vanilla FFN with low dimension of inner-layer. The FFN$_{S-MoE}$ indicates the method of sparse mixture of experts from Switch Transformer (Fedus et al., 2022) used to FFN.

| Model | Steps per second | | |
|---|---|---|---|
| Sequence Length | 128 | 1K | 4K |
| Transformer | 20.21 | 3.45 | - |
| Longformer | 12.30 | 2.44 | - |
| Performer | 25.48 | 4.17 | 1.18 |
| Flowformer | 14.29 | 4.50 | 1.16 |
| Transformer+ | 29.09 | 4.55 | - |
| Longformer+ | 20.38 | 2.70 | - |
| Performer+ | **41.22** | 6.67 | **1.85** |
| Flowformer+ | 24.93 | **7.14** | 1.82 |

Table 3: Efficiency analysis with the hidden dimension 768, layers 6, number of heads 12, inner-layer dimension 3072, and batch size 32 on byte-level text classification benchmark. "+" represents + MSCFFN. "-" indicates the out-of-memory situation.

Compared with the original FFN whose complexity is $O(8ld^2)$, MSCFFN can save more than 60 percent of the computing resources.

Ability of model representation: Multi-subspace help the model to learn different representations actively. The cross mechanism of subspace and the larger coefficient of inner-layer are beneficial to enlarge the matrice space and obtain a stronger representation capability. What's more, MSCFFN has a deeper network which is 5 compared with 2 in original FFN, and this motivates the model to learn more complex representations.

## 4 Experiments

In this section, we testify the effectiveness and generality of our proposed model MSCFFN on the dataset of Long-Range Arena (LRA).

LRA is a benchmark which contains five different tasks: Long sequence ListOps (Nangia and Bowman, 2018), Byte-level text classification (Maas et al., 2011), document retrieval (Radev et al., 2013), image classification on sequence of pixels (Krizhevsky et al., 2009), and Pathfinder (Linsley et al., 2018). Their length of sequences ranges from 1K to 16K. LRA is a suitable benchmark for testing the quality of efficient transformer variants, especially for the ability of long input sequence. We chose this dataset to verify that MSCFFN is general for different tasks and is efficient in different sequence length. We utilize directly the official method of data processing, experiment protocol and training parameters[1] to ensure the fair comparison. And we use MSCFFN directly to replace the original FFN in four base models. The experiment is conducted on 4 V100 GPUs. To ensure result reliability, we conducted five separate experiments for each task with different seed values, averaging them for the final reported result.

The experimental results are shown in Table 1. And we can see that they all achieve a better performance in average score than baselines. The

---
[1] https://github.com/google-research/long-range-arena

MSCFFN brings 1.06 on Transformer and 1.05 on Performer averaged promotion respectively. Therefore MSCFFN has a more competitive performance over the original FFN.

To further prove the effectiveness of our multi-space cross method, we conduct ablation studies whose results shown in Table 2. The FFN$_{low-di}$ indicates the vanilla FFN with low dimension of inner-layer, and we set it to $\frac{\sqrt{5}}{2}d$ to ensure that the computation complexity is close to that of MSCFNN. The FFN$_{S-MoE}$ indicates the method of sparse mixture of experts from Switch Transformer (Fedus et al., 2022) used to FFN, meanwhile we set numbert of experts to 6 and the hidden dimension of single expert to $\frac{\sqrt{5}}{2}d$. The experimental results show that directly reducing the dimension of FFN for faster speed is harmful to the effect, even if sparse moe are used.

We also conduct experiment in efficiency analysis, which is shown in Table 3. To make the results more general, we use the model parameters of common big models and test it over a larger range of length (128-> 4K). The results indicate that MSCFFN can speedup all these models both in short and long sequence. The MSCFFN can achieve at most 1.7x speedup on Flowformer and 1.6x speedup on Performer in Table 3.

## 5  Conclusions

In this paper, we propose a new FFN structure named MSCFFN, which can replace the original FFN in Transformer models and speedup model training and inference no matter how long or short the input sequence is. In MSCFFN, we reduce the computation complexity through splitting the matrix space to several small space. Benefiting from multi-space cross mechanism, a larger coefficient of inner-layer and a deeper network, MSCFFN can achieve comparable or even better performances than the vanilla FFN. The experiment results on five tasks of LRA demonstrate the effectiveness and generality of our proposed model MSCFFN. Furthermore, the proposed model is probably not only used for FFN in transformer models, but also for other model structures which contain FFN.

## Limitations

We have to note that the cross mechanism of sub-space by element-wise product will fail to a smaller space in a small probability. For example, if two spaces are orthogonal, the element-wise product

equals zero. Therefore, although it happens with low possibility, a constraint should be added to avoid it. What's more, we believe that there are more efficient methods to conduct the cross mechanism. We leave all this for future work and we will refine it more perfect theoretically and experimentally.

## Ethical Considerations

All the datasets in our experiments are already public by other researches and can be acquired easily from network. And they don't contain real personal information and will not cause any harm. To simplify the research of others, we will release the code later.

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
