# OpenReview forum: "MSCFFN: A New FFN with Multi-Space Cross to Accelerate Transformer"
_EMNLP/2023/Conference — EMNLP 2023 Findings_

### Official Review · Reviewer_i4EW · 2023-08-02

**Typos Grammar Style And Presentation Improvements:** The presentation is overall great and…
**Soundness:** 2

**Excitement:**

3: Ambivalent: It has merits (e.g., it reports state-of-the-art results, the idea is nice), but there are key weaknesses (e.g., it describes incremental work), and it can significantly benefit from another round of revision. However, I won't object to accepting it if my co-reviewers champion it.

**Paper Topic And Main Contributions:**

This paper proposed a new structure called MSCFFN to replace the feed forward network (FFN) structure in Transformers. The MSCFFN has lower computation complexity and achieved similar or higher performance than FFN in the LRA benchmark (5 tasks).

The key idea is to split the d-dimension vector into n subspaces with each being d/n dimension, then apply elementwise product to pairs of subspaces to enlarge the representation power.

**Questions For The Authors:**

In Table 2, is it number of training steps or testing steps per second?
How were those hyper-parameters m,n selected?

**Reasons To Accept:**

The proposed MSCFFN contains the cross method of subspace, which aims to keep the representation learning power while reducing the space dimension and number of parameters. Given that FFN is a widely used structure (as Transformers are), it can inspire the community to try it to reduce the computation complexity or further improve the method.

**Reasons To Reject:**

First, the contribution looks limited to me -- 60% of computations are saved theoretically, but the complexity is still in the same order -- quadratic. With the GPU acceleration, the practical speed up might be even less.

Second, the cross method of subspace is not intuitive to me. Why randomly pair the subspaces but not consider all possible pairs? Why on;y apply ReLU to one of them but not the other? Also, the author has increased the layers from 2 to 5, making it a deeper network. I feel some reasonable baselines might be needed to demonstrate the real power of MSCFFN. For example, what if we just use d/4 as the dimension but use 4 layers of them as the FFN (this also saves complexity)?

Third, the key for this kind of substitution is to achieve the same performance. I think more benchmarks are needed in addition to LRA.

**Reproducibility:**

4: Could mostly reproduce the results, but there may be some variation because of sample variance or minor variations in their interpretation of the protocol or method.

**Reviewer Confidence:**

3: Pretty sure, but there's a chance I missed something. Although I have a good feel for this area in general, I did not carefully check the paper's details, e.g., the math, experimental design, or novelty.

---

> ### Author Rebuttal · Authors · 2023-08-29
>
> Many thanks for your insightful comments.
> 1.	In Tabel 2, it is number of training steps. And we set m to 6 and n to 12 as mentioned in Section 3.3, line 234.
> 2.	For the question of “contribution looks limite”. Our MSCFFN can replace the original FFN in Transformer models and speedup model training and inference no matter how long or short the input sequence is. And It is easily to combine with other speedup methods of linear attention to achieve faster Acceleration, such as flowformer、cosformer, which can reduce the quadratic complexity of self-attention layer related to sequence length to linear complexity.  Regrettably, as of now, there exists no method to transition the quadratic complexity of the FFN into a linear complexity structure while preserving performance integrity. Nevertheless, our forthcoming endeavors will be dedicated to meticulously exploring avenues for mitigating the quadratic complexity associated with the FFN, striving for optimal reductions without compromising efficacy. This commitment underscores the focus of our future research.
> 3.	For the question of “cross method of subspace”. Firstly, Due to computational complexity, we pair subspaces, but not all possible pairs. We also experimented with all possible pairs, it exhibited superior results. But to balance the effect and computation, we finally choose to pair the subspaces. Secondly, we apply ReLU to one of them is to introduce a controlled form of non-linearity while maintaining the ability to pass through information unaltered. If non-linear functions were applied to both parts, the information might undergo extensive transformations in both branches, leading to a loss of the original input's identity. Thirdly, we did a comparative experiment without a multi-space crossover mechanism, and the average score would drop by 1. The multi-space intersection mechanism in our MSCFFN is indeed important..
> 4.	Finally, thanks for your suggestions, we can add more experimental results in the final version.

---

### Official Review · Reviewer_iZ74 · 2023-08-02

**Typos Grammar Style And Presentation Improvements:** There should be a space between "(" a…
**Soundness:** 2

**Excitement:**

2: Mediocre: This paper makes marginal contributions (vs non-contemporaneous work), so I would rather not see it in the conference.

**Missing References:**

1. Zhang, Zhengyan, et al. "Moefication: Transformer feed-forward layers are mixtures of experts." arXiv preprint arXiv:2110.01786 (2021).
2. Chelombiev, Ivan, et al. "Groupbert: Enhanced transformer architecture with efficient grouped structures." arXiv preprint arXiv:2106.05822 (2021).

Please also check the related work section in the above papers.

**Paper Topic And Main Contributions:**

The paper proposes to reduce Transformer's computational cost by reducing the computational complexity of feed forward layers, specifically, it splits the large matrix space to several small spaces and uses the Multi-Space Cross method to prevent performance decline. The paper evaluates the accuracy and efficiency on the Long-Range Arena benchmark and shows the proposed MSCFFN can achieve better results with a faster speed.

**Reasons To Accept:**

The proposed MSCFFN can save more than 60 percent of the computation resources without undercutting model performances.

**Reasons To Reject:**

The paper fails to compare to related work. Even though the goal of the proposed model is to reduce computational cost of transformers, the method itself is a method to reduce FFN, and the paper fails to introduce related work and compare to them. Please see related work in "Missing References" section.

**Reproducibility:**

3: Could reproduce the results with some difficulty. The settings of parameters are underspecified or subjectively determined; the training/evaluation data are not widely available.

**Reviewer Confidence:**

3: Pretty sure, but there's a chance I missed something. Although I have a good feel for this area in general, I did not carefully check the paper's details, e.g., the math, experimental design, or novelty.

---

> ### Author Rebuttal · Authors · 2023-08-29
>
> Thank you for your valuable feedback.
> 1.	We acknowledge your raised concerns regarding the juxtaposition of our paper with the existing body of work. The constraints imposed by temporal limitations necessitated the evaluation of merely four foundational models. It is important to highlight that our methodology distinctly diverges from the paradigms proposed by MoEﬁcation and GroupBERT. Rest assured, we are committed to not only referencing the pertinent sources but also incorporating the comparative outcomes of our experiments within the definitive iteration of our manuscript.
> 2.	We apologize for the typos and grammatical errors. Thanks for bringing these issues to our attention in detail. We will carefully review the whole paper to ensure that the overall writing meets the high standards of the EMNLP.

---

### Official Review · Reviewer_wC4i · 2023-08-10

**Soundness:** 3

**Excitement:**

4: Strong: This paper deepens the understanding of some phenomenon or lowers the barriers to an existing research direction.

**Missing References:**

None

**Paper Topic And Main Contributions:**

The authors tried to optimize the feed-forward network by splitting the huge matrix into a smaller matrix and used the multi-space cross method to improve the performance.

**Questions For The Authors:**

Question A: What will be the impact of the proposed approach on the overall parameter of the model?
Question B: why is a non-linear function applied only to one of the pairs? why can't we apply to both pairs?

**Reasons To Accept:**

The authors factorized the huge matrix and proposed multi-space cross mechanisms to optimize the feed-forward block of the transformer. The experiment show improvement in the performance when compared with the baseline.

**Reasons To Reject:**

None

**Reproducibility:**

3: Could reproduce the results with some difficulty. The settings of parameters are underspecified or subjectively determined; the training/evaluation data are not widely available.

**Reviewer Confidence:**

4: Quite sure. I tried to check the important points carefully. It's unlikely, though conceivable, that I missed something that should affect my ratings.

**Typos Grammar Style And Presentation Improvements:**

It would be great if the authors do one additional round of proof reading. There are a few minor things like Early exiting -> early exiting.

---

> ### Author Rebuttal · Authors · 2023-08-29
>
> Many thanks for your insightful comments.
> 1.	For Question A: For the number of parameters for different models, the FFN has a parameter value of $8d^2$, whereas our MSCFFN has a parameter value of $2.25d^2$. MSCFFN has a smaller number of parameters and actual GPU memory usage compared to FFN, giving it an advantage in model size.
> 2.	For Question B: The primary reason for non-linear functions only to one of the pairs is to introduce a controlled form of non-linearity while maintaining the ability to pass through information unaltered. If non-linear functions were applied to both parts, the information might undergo extensive transformations in both branches, leading to a loss of the original input's identity.
> 3.	We apologize for the typos and grammatical errors. Thanks for bringing these issues to our attention in detail. We will carefully review the whole paper to ensure that the overall writing meets the high standards of the EMNLP.

---

### Meta-Review · Area_Chair_ASxk · 2023-09-15

**Recommendation:** 2

**Metareview:**

This paper aims to reduce the cost of MLPs found in Transformer architectures by decomposing the fully connected layers into smaller parallel components. Results demonstrate improved performance/faster runtime. Reviewers mostly agree on the importance of this research direction and find the proposed method promising. However they point out the lack of comparison with related work (sparse-MoEs) and ablations; which impacts the soundness of the claims made in the paper.

---

### Decision · Program_Chairs · 2023-10-07

**Decision:**

Accept-Findings

**Comment:**

This paper aims to reduce the cost of MLPs found in Transformer architectures by decomposing the fully connected layers into smaller parallel components. Results demonstrate improved performance/faster runtime. Reviewers mostly agree on the importance of this research direction and find the proposed method promising. However they point out the lack of comparison with related work (sparse-MoEs) and ablations; which impacts the soundness of the claims made in the paper.